# Focused Update on Pulmonary Hypertension in Children—Selected Topics of Interest for the Adult Cardiologist

**DOI:** 10.3390/medicina56090420

**Published:** 2020-08-19

**Authors:** Sulaima Albinni, Manfred Marx, Irene M. Lang

**Affiliations:** 1Paediatric Heart Centre Vienna, Department of Paediatrics and Adolescent Medicine, Medical University of Vienna, 1090 Wien, Austria; manfred.marx@meduniwien.ac.at; 2AKH-Vienna, Department of Cardiology, Medical University of Vienna, 1090 Wien, Austria; irene.lang@meduniwien.ac.at

**Keywords:** pulmonary hypertension, Fontan physiology, Scimitar Syndrome, sickle cell disease, children, segmental hypertension, CHD, heart transplantation

## Abstract

Pulmonary hypertensive vascular disease (PHVD), and pulmonary hypertension (PH), which is a broader term, are severe conditions associated with high morbidity and mortality at all ages. Treatment guidelines in childhood are widely adopted from adult data and experience, though big differences may exist regarding aetiology, concomitant conditions and presentation. Over the past few years, paediatric aspects have been incorporated into the common guidelines, which currently address both children and adults with pulmonary hypertension (PH). There are multiple facets of PH in the context of cardiac conditions in childhood. Apart from Eisenmenger syndrome (ES), the broad spectrum of congenital heart disease (CHD) comprises PH in failing Fontan physiology, as well as segmental PH. In this review we provide current data and novel aspects on the pathophysiological background and individual management concepts of these conditions. Moreover, we focus on paediatric left heart failure with PH and its challenging issues, including end stage treatment options, such as mechanical support and paediatric transplantation. PH in the context of rare congenital disorders, such as Scimitar Syndrome and sickle cell disease is discussed. Based on current data, we provide an overview on multiple underlying mechanisms of PH involved in these conditions, and different management strategies in children and adulthood. In addition, we summarize the paediatric aspects and the pros and cons of the recently updated definitions of PH. This review provides deeper insights into some challenging conditions of paediatric PH in order to improve current knowledge and care for children and young adults.

## 1. Introduction

Pulmonary Hypertension (PH) or Pulmonary Hypertensive Vascular Disease (PHVD) is a severe condition characterized by a progressive increase in pressure or vascular resistance, leading to chronic right heart failure. While treatment guidelines in childhood are mainly based on adult data and experience, significant differences exist regarding aetiology, concomitant conditions and presentation. Current registries have provided a growing understanding of specific features, presentation, epidemiology and outcomes of paediatric pulmonary hypertension (PH) [1,2,3,4]. In contrast to adults, in whom left heart disease accounts for 40–70% of PH aetiologies [5], childhood PH is very often linked to developmental and lung growth disorders as well as congenital heart disease and genetic syndromes [1,6,7]. In many cases, especially in smaller children, PH is based on multifactorial mechanisms.

The aim of this review is to discuss some of the novel changes of the sixth WSPH, and to highlight some challenging issues specific for paediatric PHVD that have not been covered by previous reviews. Furthermore, we discuss advanced PH treatments due to left heart disease in children. Some conditions are still not covered in current guidelines and recommendations and require a patient-based approach. 

## 2. Definition and Classification

Since the first WSPH in 1973, PH has been defined as mPAP > 25 mmHg at rest, measured by cardiac catheterisation. Basically, this definition has been applied equally to adult and paediatric PH, though some differences exist due to specific characteristics of paediatric vasculature. In 2018, working members of the sixth WSPH introduced an updated classification on PH (Table 1) and proposed a revised definition of PH, setting the cut off of mPAP to >20 mmHg [8]. This recommendation created controversy between PH experts. The basic argument for changing the original definition was the fact that the first cut off of mPAP > 25 mmHg was a historical and pragmatic choice introduced during the first WSPH in order to distinguish rare and severe forms of primary PH, with awareness that mPAP is normally not higher than 15 mmHg in healthy persons. Since then, available hemodynamic data in healthy individuals showed that the normal range of mPAP is approximately 14 ± 3.3 mmHg and that the upper limit (>97.5th percentile) of normal is 20 mmHg [9]. The term borderline PH was an attempt to address this gap between the upper limit of normal and 25 mmHg. However, it did not find unanimous acceptance by leaders of the WSPH working groups. Furthermore, a growing number of studies over the past 5 years showed that a measured mPAP of >20 mmHg was associated with increased mortality, independent of underlying disease. Increased risk was mostly reported for left heart disease, COPD and systemic sclerosis [10,11,12,13,14]. Apart from that, symptomatic CTEPH patients with mPAP levels between 20 and 24 mmHg showed significant improvements after interventional balloon pulmonary angioplasty or surgical endarterectomy, indicating a possible clinical relevance of these pressure levels [15,16].

Based on these findings, the sixth WSPH concluded that mPAP values between >20 mmHg and 24 mmHg represent a hemodynamic risk factor and that such patients should receive close monitoring and specialized management. However, whether or not such patients would benefit from treatment remains unclear. 

For children, the new threshold of definition does not implicate a substantial clinical advantage in management. First, natural variations of hemodynamics are observed during childhood. In the foetal period, pulmonary artery pressure is as high as systemic pressure and falls to normal levels within the first 6 to 8 weeks of life. Therefore, it is recommended to use this definition only after 3 months of life. Apart from that, patients with congenital heart disease such as large intra- or extra-cardiac shunts, have elevated pressures due to increased pulmonary blood flow, mostly with normal PVR. Therefore, working members of the first paediatric PVRI taskforce introduced the term PHVD (pulmonary hypertensive vascular disease) and recommended the introduction of the indexed PVR to characterize these patients [7]. Patients with precapillary PH and elevated PVRI—indicating the existence of PHVD—would benefit from targeted therapies, whereas in flow mediated PAH with normal PVRI simple shunt closure is the first choice of treatment.

In many other cases, especially in infancy, mildly elevated PA pressures will resolve without targeted therapies during the first months of life (PPHN, BPD) [18]. As a consequence, not all of them will undergo complete hemodynamic evaluation, which is somehow more complicated in childhood as it warrants at least conscious sedation. Labelling these patients with PA pressures between 20 and 24 mmHg as PAH would create a problem not only for the patients and their caregivers—as it means a dramatic psychological burden for the families—but also for physicians, because targeted therapies especially are not investigated yet in this range of PA pressures and, therefore, are not approved. Furthermore, it seems questionable whether asymptomatic patients with PA pressures between 20 and 24 mmHg should be treated, given the high costs and potential side effects of targeted therapies. Based on data of current registries, children with PAH present with far higher PA pressures than 20 mmHg, indicating that patients become symptomatic very late; therefore, it is questionable whether lowering the threshold would help to detect early disease [1,2,19,20]. On the other hand, normal pulmonary artery pressure does not necessarily exclude vascular disease in special conditions of congenital heart disease, such as Fontan physiology. Due to the low flow non-pulsatile nature of the special hemodynamic condition in this growing number of patients, mPAP is often low despite pulmonary vascular disease. In the Panama paper, an indexed PVRI > 3 WU × m^2^ or a transpulmonary gradient (TPG) of >6 mmHg was suggested to identify those patients who might benefit from targeted pharmacological treatment [6,7,21,22,23].

The sixth WSPH was suggested to incorporate PVR in the definition of precapillary PH. The use of PVR should help to distinguish between isolated precapillary, combined pre- and postcapillary and postcapillary PH. The application of PAWP has not been changed to define postcapillary PH; the presence of PVR ≥ 3 WU, together with a PAWP > 15 mmHg, suggests combined pre- and postcapillary PH [8].

The paediatric task force decided to accept this definition for children, but reinforced the recommendation to index PVR [6]. The updated definitions of paediatric PH proposed by the European Paediatric Pulmonary Vascular Disease Network (EPPVDN) [17,24] are listed in Table 1. The 2019 revision of the clinical classification is included for comparison.

For patients at risk of developing relevant PH (such as family members of hereditary PAH or patients with systemic sclerosis), regular screening has been recommended to detect early disease [25], although some entities, such as systemic sclerosis and CTEPH, are very rare in childhood [1,2,23]. However, close monitoring of patients with invasively measured PA pressures between 20 and 24 mmHg is reasonable. 

## 3. Group 1 PH (Table 1)

### 3.1. Vasoreactivity

In all patients with IPAH/FPAH, hemodynamic assessment should include acute vasoreactivity testing (AVT) to identify patients who may benefit from long term calcium channel blocker (CCB) therapy. A positive AVT in adult patients is defined by a decrease in mPAP by at least 10 mmHg to a value of <40 mmHg with unchanged cardiac output (Sitbon criteria) [26]. As in adults, the preferred testing agent in children is inhaled NO at a concentration of 10–80 ppm. Based on recently published data, including children and adults, a decrease in mPAP by at least 10 mmHg to a value of <40 mmHg with unchanged cardiac output were best to identify subjects who will benefit from long term CCB therapy and carry a better prognosis; therefore, it is recommended to also use these criteria in children [27]. Barst criteria with AVT definition of a change in mPAP of 20% (with increase or no change in CI and a decrease or no change in the pulmonary vascular resistance to systemic vascular resistance ratio) of baseline were not predictive of long-term survival and are therefore mostly replaced by the guideline recommendations [8,28].

Recently published data on AVT in children have demonstrated that only 10–15% of children are responders by this definition, which is in keeping with adult data [2,20]. Only half of the adult responders show sustained long term hemodynamic and clinical improvement on CCB therapy; therefore, close monitoring is recommended in all patients on CCB therapy [6]. Substances such as nifedipine, amlodipine and diltiazem have been used in children. Due to the possible negative inotropic effects, CCB are contraindicated in infants <1 year of age [23].

In children with CHD associated PAH, vasoreactivity testing is performed to decide on operability in borderline cases. Patients with post-tricuspid shunts, who show a decrease in PVRI to 6 WU × m^2^ or a PVR/SVR < 0.3 on AVT are considered eligible for surgical repair. However, AVT is only one of the determinants of operability; other factors, such as age, comorbidities, type of lesion and saturation levels are all associated with a reversal of PAH after surgical repair and should also be considered for decision making [29].

### 3.2. Advanced Treatments for Patients Who Are Not Vasoreactive

In children with a negative response to AVT, the paediatric taskforce recommends the use of targeted therapies after individual risk stratification. Similar to adults, a higher risk in children includes hemodynamic measures such as RAP > 10 mmHg, PVRI > 20 WU × m^2^ and mPAP/sPAP ratio > 0.75 [6,30]. Recently, the working group members of the EPPVDN have updated and summarized all determinants of risk in children. In addition to hemodynamic measures, clinical and echocardiographic evidence of right heart failure, progression of symptoms, WHO functional class III and IV, failure to thrive, syncope and elevated or increasing BNP/NT–pro BNP indicate therapy escalation [17]. In patients with low risk, the recommended agents are endothelin receptor antagonists (ERAs), such as bosentan and ambrisentan and/or phosphodiesterase type 5 inhibitors (PDE-5i), such as sildenafil and tadalafil. The early combination of these substances seems to be reasonable. For patients remaining at low risk, the additional inhalation of iloprost or treprostinil can be beneficial. Intravenous or subcutaneous prostacyclin analogues should be offered early to patients with high risk PAH [6,8]. The use of novel oral agents, such as selexipag and macitentan in children is currently under investigation. Background therapy with diuretics, anticoagulation and digoxin should be initiated on an individual basis. Overall, current recommendations on targeted therapies in children are mainly based on adult data and experience, and larger randomized trials in children are lacking due to the relatively small numbers and the large heterogeneity in paediatric PH. Data on the use of targeted therapies in paediatric specific conditions associated with PH are discussed in the following chapters. 

#### End-Stage/Bridging Treatment Strategies

Progress in medical treatment has improved quality of life and the survival of PH patients over the last few decades. However, in patients who continue to deteriorate despite maximal medical therapy, lung transplantation remains the ultimate treatment option. Various therapeutic strategies have been proposed to ameliorate right heart function, low systemic cardiac output and syncope, the latter of which often presents in children with suprasystemic pressures. The concept behind these procedures is based on the observation that, in contrast to IPAH, Eisenmenger patients often have better right ventricular function. These procedures create similar relief as that present in Eisenmenger patients.

Balloon atrial septostomy (BAS) is a percutaneous procedure by which an atrial communication is created via balloon dilation of the atrial septum. Resulting right-to-left atrial-level shunting allows the decompression of the right heart with increased left ventricular pre-load and cardiac output at the expense of cyanosis. Indications for BAS include syncope, right heart failure, refractory to chronic PAH-targeted therapy, and symptomatic low cardiac output states [31,32]. In the hand of experienced interventionists, it has been a relatively safe procedure with beneficial hemodynamic effect in children and adults with severe PAH [33,34,35]. The risk factors associated with high procedural mortality were a mean right atrial pressure > 20 mmHg, resting arterial oxygen saturation < 90%, a left ventricular end-diastolic pressure > 18 mmHg, an indexed pulmonary vascular resistance > 55 Wood units·m^2^ and severe right ventricular failure [34,36,37]. In general, RAP > 20 mmHg and a resting oxygen saturation < 90% in room air are considered as contraindication for this procedure [6,38].

A relatively novel strategy is Potts shunt (PS), which is a surgically created connection of the left pulmonary artery to the descending aorta [39,40,41]. It results in the undersaturation of the lower body; however, compared to BAS it has the advantage of providing high oxygen saturated blood for coronary arteries and the central nervous system. PS decreases RV afterload in systole and in part in diastole, by that improving interventricular interactions and LV ejection fraction. This approach has been proposed as a bridge or even as an alternative to lung transplantation [39,41].

In the case of a persistent patent ductus arteriosus (PDA) interventional stent implantation is considered the safer strategy [42,43]. Stenting of the duct has been performed in infants with various complex cardiac lesions with duct-dependent physiology as palliative procedure [35,43,44,45]. The so-called functional Potts shunt has also been used in various forms of PH, including postcapillary PH [43]. In general, Potts shunt has been mainly used in children, though the first experience in young adults has been published [42,46,47]. In adults, the interventional Potts shunt is a high risk procedure [48].

Patients not responding to treatment should be considered for lung transplantation. Based on ISHLT registry data, IPAH is the second most common indication for paediatric lung transplantation (LTX), where DLTX is the treatment of choice. The median survival after lung transplantation was 5.7 years, compared to adults with 6.4 years [49,50].

Studies comparing the outcomes of PAH patients after DLTX and HLTX showed similar results regarding survival [51,52]. In general, the proposed cut off values for successful DLTX are LVEF between 32% and 55% and RVEF between 10% and 25%. For any lower value, HLTX might be the safer procedure. Nevertheless, by now, the main indication for HLTX in PAH is still the complex Eisenmenger syndrome [50,51].

### 3.3. Pulmonary Hypertension in Fontan Physiology

Fontan circulation is a surgically created low flow hemodynamic condition to palliate patients with univentricular physiology, such as hypoplastic left heart syndrome, tricuspid atresia, RV hypoplasia, and many other complex heart defects sharing the common feature that the ventricular cavity cannot be divided in two functional pumps. The Fontan procedure in such patients is normally performed after a number of surgical and interventional steps and results in a condition in which the systemic venous blood enters the lung in a non-pulsatile manner by bypassing the heart. Fontan physiology is far away from being natural, though this kind of palliation has saved the lives of many infants born with univentricular hearts. Progress in surgical techniques’ medical management and novel interventional procedures has dramatically improved the survival of Fontan patients. Though still being a palliation, up to 70% of patients will suffer from Fontan failure until the third decade of life [53,54,55]. This is where pulmonary hypertension comes to play a role as, among other factors (i.e., arrhythmias, ventricular failure), abnormal pulmonary hemodynamics are one of the main causes for failing Fontan [56,57,58].

Due to reduced pulmonary blood flow and low cardiac output, resulting in low PA pressures in Fontan circulation, the conventional definition of PH does not apply. Therefore, the first paediatric PH taskforce introduced a special definition of abnormal pulmonary hemodynamics in Fontan physiology, defined by a TPG > 6 mmHg and a PVRI > 3 WU × m^2^, indicating pulmonary vascular disease in this population [7,17].

An abnormal pulmonary vascular bed is a hallmark of patients with univentricular hearts. Pulmonary arteries are hypoplastic, stenosed or have abnormal arborization early in life. Throughout life they are subjected to several alterations that are likely to affect their further growth. Interventions and surgical procedures may result in distortions, stenoses and abnormal pulmonary blood flow. However, the development of the pulmonary vascular bed normally occurs early in life when pulmonary blood flow is pulsatile. Pulsatility is lost by the time of Glenn shunt and the following Fontan procedure. Venous flow patterns attenuate vascular growth and the recruitment of capillaries, and contribute to abnormal pulmonary architecture [59]. There exists strong evidence for pulmonary vascular remodelling on a molecular basis. The link between non-pulsatile flow, leading to the lack of shear stress and endothelial dysfunction, has been illustrated in several experimental studies [57,59,60,61,62]. These findings are supported by studies on patients with Fontan physiology, showing the decreased expression of endothelial-derived factors, the upregulation of nitric oxide synthase, endothelin-1, and endothelin-receptors, as well as the reduced expression of bone morphogenetic protein receptor type 2—all features of advanced pulmonary vascular remodelling [60,63,64,65,66]. Interestingly, histomorphological findings from post mortem analysis of Fontan patients suggests an unusual phenotype of vascular remodelling with a thin medial smooth muscle cell layer and loss of smooth muscle cells due to apoptosis and eccentric intimal fibrosis [67]. The surrounding mechanisms contributing to vascular remodelling are complex and not yet fully understood, however the presence of pulmonary vascular disease and the consecutive elevation of PVR is one of the main drivers leading to a failing Fontan circulation [56,59].

Based on these data, there is a strong rationale for the use of targeted therapies that have been approved for the treatment of PAH. In the past few years, many studies have investigated the use of vasoactive substances in these patients. The benefit of iNO on pulmonary vascular dysfunction in the early postoperative period was clearly demonstrated by Goldman et al. more than 20 years ago [68]. Since then, a number of studies and case reports were published supporting these effects in patients after the Glenn shunt and Fontan procedure [57,58,69,70]. The effect of NO on PVR was investigated in 2003 in patients late after Fontan, showing a decrease in PVR, especially in those with elevated levels (>2 WU) [71]. An interesting study published in 2016 investigated the effects of iNO on flow patterns assessed by MRI. In this study NO led to a decreased aorta to pulmonary collateral flow and to a net increase in systemic blood flow, suggesting an improvement in cardiac output [72]. The NO pathway was first investigated in 2008 by the use of sildenafil. In this study, a single dose of sildenafil showed an increased cardiac index at rest and during exercise, and improvement in peak oxygen uptake (VO2) [73]. Sildenafil was associated with improved exercise capacity in several other studies, including paediatric and adult Fontan patients [74,75,76,77]. In selected failing Fontan patients, sildenafil has been reported to improve the symptoms and features of Fontan failure, such as plastic bronchitis [74,78]. These beneficial effects are most likely attributed to a reduction in PVR than an improvement of ventricular function [79]. Although published data on Sildenafil are encouraging, larger clinical trials to provide sufficient evidence for the use of this substance are currently lacking. Of note, there is only one "longer" term randomized, double blind, cross over trial investigating the effects of sildenafil over 6 weeks [76]. The SV INHIBITION study (NCT03997097), a French multicentre, randomized, double blind, placebo-controlled Phase III study was designed and is currently starting enrolment [80]. Results of this important trial will hopefully answer the question whether PDE5 should be routinely used in suitable Fontan patients. Increased endothelin levels and the upregulation of its receptors have been reported in Fontan patients. Thus, blocking the endothelin pathway seems to be another promising treatment target. Given the potential of hepatic failure in Fontan patients, endothelin receptor antagonists (ERAs) may raise some concerns. The safety of bosentan has been investigated in smaller studies and did not show significant adverse effects on liver function [81]. The effects of bosentan over a treatment period of 14 weeks was assessed by the TEMPO trial, a randomized placebo controlled study which showed the improvement of peak oxygen uptake and a reduction in BNP [82]. However, most patients were in NYHA functional class 1. Improvements of 6 MWD and NYHA functional classes as well as MRI derived CI have been described with bosentan in smaller studies [81,83,84]. On the other hand, an open label trial including 42 adults treated with bosentan over 6 months turned out to be negative with regard to the primary endpoint exercise capacity [84]. One recently published study, including children and adults, assessed the effects on PVR and showed a significant reduction with bosentan and macitentan [85]. The effects of macitentan in this population is currently under investigation in a prospective, multi-centre, double-blind, randomized, placebo-controlled trial (RUBATO study NCT03153137), which will include more than 100 Fontan patients beyond the age of 12 years. Its results are expected next year and will hopefully shed light on the usefulness of ERAs in this population.

The role of prostacyclines has been investigated in only two trials, which found beneficial effects of inhaled iloprost on symptoms and exercise performance [86,87].

In conclusion, there is a strong rationale for the use of targeted therapies and growing evidence for its positive impact on PVR, which is a main determinant of survival in Fontan patients. Larger randomized studies are coming up to provide sufficient evidence. However, it remains to be clarified which Fontan patients will benefit the most, which substances are the most appropriate and which determinants and endpoints should be assessed when using these agents. Although PVR is the most important determinant, its calculation by catheter alone may be difficult due to inaccuracies of pulmonary blood flow measurements. The combination of MRI and catheter seems be the ideal strategy for correct PVR calculation [88,89,90]. However, currently this method is not yet established in all centres.

### 3.4. Eisenmenger Syndrome

Within the group of PAH associated with CHD (APAH-CHD), PAH in Eisenmenger Syndrome (ES) represents the most severe phenotype of disease. It is defined by any large intra or extracardiac defect, permitting unrestricted pulmonary blood flow and the transmission of systemic pressure to the lung, which results in a balanced or predominant right-to-left shunt caused by a markedly elevated pulmonary vascular resistance [91]. Associated hemodynamic conditions include shunts at atrial or ventricular levels or both (AVSD) or at arterial level (aortopulmonary window, truncus arteriosus communis TAC), as well as complex heart defects, such as discordant ventriculo–arterial connection and single ventricle physiology with unrestricted pulmonary blood flow (DILV, DORV). The risk of developing ES is determined by the type of CHD, the size of the defect and the associated comorbidities, such genetic factors or congenital syndromes [92,93,94,95]. Though more prevalent in adulthood, some underlying complex forms of CHD (TAC, DILV, DORV) can promote ES very early in childhood with increasing risk in the presence of genetical comorbidity, such as Down syndrome [6,96].

According to different PAH registries, adult ES carries a prevalence between 1 and 12% in patients with CHD [97,98,99,100].

In children, APAH-CHD—the largest group of PAH—accounts for 24–75% of cases [2,3,28,101]. The prevalence of early ES remains unclear. Data from the UK pulmonary hypertension service for children revealed that 31% of APAH-CHD cases were ES [102].

However, as a consequence of improved diagnostics and early surgical management, ES has become less prevalent over time and is increasingly replaced by PAH-CHD occurring after or despite repair.

The clinical hallmark of ES at all ages is cyanosis and secondary erythrocytosis with ensuing multisystemic organ involvement responsible for the high morbidity seen in those patients. Quality of life and functional impairment are mainly driven by the degree of cyanosis. Therefore, the management of ES patients requires a lifelong monitoring within a tertiary centre where the variety of multiorgan complications is addressed. In this context recommendations for paediatric patients have been widely adapted from adult data [97,103,104]. Iron deficiency is an important frequently underrecognized phenomenon and needs aggressive treatment.

Regular phlebotomies for hyperviscosity syndrome have been abandoned due to resulting iron deficient anaemia, which is a risk factor of cerebrovascular events. ES associated with haematological abnormalities promote a complex paradox with increased risk of bleeding and thrombosis. Currently, there is no recommendation for anticoagulation treatment, except for special cases of arrhythmias and thromboembolic events, which are very rare in paediatric ES [97,105,106].

Chronic oxygen therapy is similarly debatable and reserved for selected patients with exercise-induced cyanosis [107]. Non cardiac surgery in ES carries a significant risk of morbidity and mortality, due to possible risk of bleeding, thrombosis and arrhythmias. Hemodynamic instability due to hypotension aggravating hypoxemia, and fluid shifts during surgery, need careful consideration with regard to the risk-to-benefit ratio.

The hypoxia induced alteration of the immune system and brain barrier subjects patients with right-to-left shunting to increased risk for cerebral abscesses and endocarditis [108,109]. Furthermore, renal impairment with abnormal fluid retention and hyperuricamia are common findings in ES patients and independent risk factors of death. Other, less serious, conditions include gall stones, cholecystitis or scoliosis.

The thoughtful consideration and management of all of these complications, together with the use of advanced specific PAH drugs, have improved the management and survival of ES patients. Currently there is established evidence for the application of targeted therapies in ES, though data are mostly based on adult studies. Several studies, including paediatric patients, have demonstrated the beneficial effects of sildenafil and bosentan on 6 MWD, WHO functional class and outcome [110,111,112,113,114,115]. In adults, a single or dual PAH specific therapy with Sildenafil and/or bosentan was associated with a significant reduction in death or transplantation [116,117].

The effect of macitentan, the second-generation ERA was studied in the randomized multicentre MAESTRO trial including paediatric ES patients > 12 years of age. Unfortunately, it did not show the superiority of macitentan compared to the placebo on 6 MWD and WHO class in this population [118].

Prostacyclines are highly effective in the treatment of PAH at any age, but only a few studies exist on its use in ES patients. Epoprostenol improved functional class 6 MWD and oxygen saturation in a retrospective study in severely ill patients [119,120]. However intravenous prostacyclin carries the drawback of increased risk of infection and embolism in this population. Subcutaneous treprostinil use in ES was first published in 2018 and demonstrated the beneficial effects on WHO class, 6 MWD and BNP levels in a relative poor group of APAH-CHD mainly ES patients [121]. The role of Selexipag, an oral PGI agonist, in ES patients is yet to be determined.

Overall, the multidisciplinary approach to ES patients, including supportive management and treatment with targeted therapies, has improved quality of life, morbidity and outcomes, yet mortality is high in ES patients, especially in young patients with complex CHD [97,122].

In contrast to previous reports demonstrating better survival in ES patients compared to other PAH forms, recently published data report high mortality that is comparable to IPAH and FPAH [123,124,125]. This is also true for childhood ES. Van Loon et al. demonstrated a median survival of 11.4 years in paediatric ES patients, compared to reported survival rates of 34–40 years in adults [2,97,124]. These conflicting results can be explained by the effect of inclusion bias of studies, as only ”survivors” of an adult age could enter clinical follow up at research institutions. Therefore, outcome data in ES patients should be interpreted with caution. Interestingly, when looking at the survival of adult ES patients, those with pretricuspid shunts had significantly worse outcomes [124,125]. This is mainly explained by the loss of the foetal phenotype of RV, which is increasingly subjected to volume overload and is less capable of maintaining high pressures. Heart failure has recently been recognized as the most common cause of death (34%) in adult ES patients, followed by infection (26%) and sudden cardiac death 10% [124]. The ultimate treatment option for severely ill ES patients is either combined heart–lung transplantation (HLTx) or lung transplantation (LTx) with concomitant cardiac repair, though the right timing is difficult, and many patients may not be eligible due to underlying comorbidities.

In summary, ES patients remain a complex and challenging group in APAH-CHD, requiring a personalized approach and thoughtful consideration throughout life. Despite the decreasing prevalence, due to the increasing complexity of patients, of whom not all will be suitable for surgical repair, ES is not likely to disappear.

#### Down Syndrome and PHVD

Trisomy 21 (T21) or Down Syndrome is currently recognized as an independent risk factor of pulmonary hypertensive vascular disease outside CHD. The incidence of Down Syndrome is approximately 1 in 670–700 children. Up to 50% of patients with Down Syndrome have CHD, with the most common lesion being AVSD (60%) followed by VSD (20%) [126,127]. Infants with AVSD tend to develop earlier PAH and are more likely to present with postoperative PH crisis [128]. Even in the absence of CHD new-borns with T21 are at higher risk of developing PPHN (persistent pulmonary hypertension of the new-born), at an estimated rate of 1.5–5.2% [129].

Mechanisms involved in the higher susceptibility to develop more severe PAH include several respiratory abnormalities. T21 typical anatomical features, such as macroglossia, mid-facial hypoplasia, chronic upper respiratory tract infections, enlarged adenoids and reduced muscle tone may result in chronic hypoxia, hypercapnia and inflammation [128], which are all known triggers of pulmonary vascular disease. On the other hand there is evidence of lung hypoplasia, decreased alveolarisation and decreased pulmonary vascular growth, most probably as a result of the overexpression of antiangiogenetic genes (endostatin, regulator of calcinuerin-1 and ß-amyloid peptide) present on chromosome 21 [130,131,132]. Genetically determined lung hypoplasia together with the typical features of T21 with or without CHD predispose affected children to a higher risk of developing PH (Table 1).

Down Syndrome related PH has therefore been recognized as a developmental lung disease, with further risk of PH in the context of additional respiratory conditions or CHD. It has therefore been categorized mainly into PH group 3, or into groups 1 or 2 in the presence of CHD.

## 4. Group 2 PH (Table 1)

Left heart failure (HF) in children is a complex syndrome with heterogeneous aetiology and presentation. The majority of heart failure cases in adults are due to ischemic heart disease, which is an infrequent cause among paediatric patients. In these, numbers include cardiomyopathy, with an annual incidence estimated at 1.13 cases per 100,000 children and children with CHD [133,134]. Two single site studies have indicated that more than half of paediatric HF cases were in children with CHD [134,135,136]. Currently, there are limited data on the prevalence and implications of pulmonary hypertension in paediatric patients with advanced left heart failure. According to the PH registry in the Netherlands, PH due to left heart disease accounts for 5% of all paediatric PH patients with most of them suffering from CHD [2]. While postcapillary PH due to mitral stenosis or aortic valve stenosis is surgically treated, diastolic left heart dysfunction is an important cause for late PH in older children or young adults with CHD. This may occur especially after the repair of coarctation, aortic stenosis, in patients with Shone complex or with hypoplastic left heart Syndrome (HLHS). In addition, complex congenital heart defects, such as failing univentricular hearts (either of left or right ventricular morphology) may also present a clinical phenotype of left heart failure.

Patients with heart failure due to congenital or acquired cardiomyopathies with or without PH receive pharmacological treatment, basically similar to management strategies in adults including diuretic agents, such as furosemide and spironolactone, angiotensin-converting enzyme inhibition, digoxin and beta blockers [5,137].

Regarding PH-specific therapy in general, its use is not recommended [138]. In CpcPH (combined post- and pre-capillary PH) due to either HFpEF or HFrEF, macitentan showed a signal of increased fluid retention versus a placebo [139]. Although most trials to this date have been negative, PH-specific therapy in those with hemodynamic evidence of severe pulmonary vascular disease may be applied in highly selected patients.

Furthermore, atrial septostomy [140,141], reversed Potts shunt [43,46,142] and pulmonary artery banding [143] were proposed as palliative procedures in children refractory to medical treatment to improve symptoms and quality of life. Among these procedures, the creation of interatrial shunting has recently been investigated in adults with heart failure in large studies. Atrial septostomy with the implantation of an interatrial shunting device (IASD) resulted in left atrial decompression and the reduction in pulmonary artery wedge pressure (PAWP), the procedure was safe and associated with a significant improvement of symptoms, WHO functional class and 6 min walking distance in patients with HFpEF and HFrEF [140,144,145]. In decompensated heart failure, mechanical support with or without atrial septostomy [143,146,147] and finally transplantation are the ultimate treatment options.

### 4.1. Pulmonary Hypertension Due to Left Heart Disease: The Role of Mechanical Support—The Way to Transplantation

High PVRI (>6 WU × m^2^) is considered a contraindication for transplant; however, the use of VADs has been shown to decrease PVR [148]. The use of paediatric VADs has increased dramatically since 2005, with 20% of paediatric patients being bridged to transplant with a device [149]. The HeartWare^®^ assist system is a therapeutic option for larger children and adolescents. In new-borns and small children, the paediatric Berlin Heart Excor^®^ may be applied, even in small infants with a body weight of less than 3 kg, whereas for the HeartWare VAD, 17 to 20 kg seems to be the lowest range due to the limitation of flow to a minimum of 2 L/min [150]. In a most recent study analysing the United Network for Organ Sharing (UNOS) database, mechanical circulatory support (MCS) (LVAD, RVAD, BiVAD, or TAH, respectively) was used in 20% overall (n = 426), with 57% of those with PVRI < 3, 27% with PVRI 3–6, and 16% with PVRI > 6 WU × m^2^. Patients supported with MCS (median waitlist duration about 3 months) had a significantly higher chance of a positive waitlist outcome than those without such support regardless of PVRI. This was most pronounced with a PVRI greater than 6 WU × m^2^ [151]. The role of PVR was analysed in a large multicentre paediatric heart transplantation study, published by Richmond et al. in 2015. In this study, 795 paediatric heart transplant recipients without congenital heart disease were investigated regarding early and intermediate outcomes using a cut off PVRI value of 5 WU × m^2^ (602 pts with PVRI < 5 WU × m^2^, 193 pts with PVRI > 5 WU × m^2^). Preoperative elevated PVRI was not associated with increased mortality after transplantation, though patients with elevated PVRI were more likely to receive intensive medical therapy, such as nitroprusside or inhaled NO. The authors conclude that children—in contrast to adults—may allow for faster reverse remodelling of the pulmonary vascular bed and that younger donor hearts may do better with increased afterload after transplantation and that elevated PVR does not necessarily contradict paediatric heart transplantation [152]. While the elevation of PVRI precludes heart transplantation in adults, these results in children may encourage individual decision making in the presence of PVRI elevation. However, further studies will be necessary to verify these findings.

The mechanism for the development of PH in left heart failure is mainly based on the dysregulation of vascular smooth muscle tone and structural remodelling in the pulmonary arteries and/or pulmonary veins. While the former is reversible over a short time period using vasodilators, the latter is less responsive to acute intervention and may resolve only with chronic therapy before or in response to heart transplantation. Pulmonary venous congestion in heart failure (HF) may lead to the predominant remodelling of pulmonary veins and the severity of venous remodelling might be associated with the severity of PH in HF. Fayyaz et al. found that more venous intimal thickening was present compared with arterial intimal thickening in those with PH-LHD, and this was similar to changes seen in people with PVOD [153]. The reverse-remodelling of the venous vessels takes time, which in part might explain the beneficial aspect of MCS running over months, leading to a 50% reduction in waitlist mortality in the new era [154] (example shown in Figure 1).

### 4.2. Pulmonary Hypertension Due to Left Heart Disease and Subsequent Heart Alone Versus Heart and Lung Transplantation

Orthotopic heart transplantation (OHT) is still considered the definitive treatment for end-stage HFrEF. The most common indications for OHT in children include end-stage congenital heart disease and cardiomyopathy [155,156,157]. Unfortunately, patients with PH-LHD have worse outcomes post-transplantation. Elevated pre-transplant PVR ≥ 3.75 Wood Units is associated with increased 30 day post-transplant mortality (25.5 vs. 6.4%) [158]. Thus, listing patients before the deterioration of quality of life due to progressive decline in cardiopulmonary function and before frequent hospital admissions is important.

The timing of transplantation must be guided by the natural history of the disease, which varies among patients with CHD (with or without ES) and cardiomyopathy, mainly RCM. Several reports from large institutions and transplantation registries have identified congenital heart disease as a risk factor for mortality after heart transplantation in children. Moreover, infants with previously palliated HLHS were shown to have the worst outcomes. Everitt et al. showed that 1-year survival after heart transplantation in 739 infants 6 months old or younger was the lowest for infants with previously palliated HLHS (70%) compared with 89% in those with cardiomyopathy [159]. At 10 years following transplantation, 40% had died and 7% had undergone re-transplantation [160]. One can speculate that HLHS patients transplanted with prior Glenn or Fontan circulation had worse outcomes due to the underestimated elevation of pulmonary vascular resistance (PVR). In these patients, haemodynamic catheter measurements can be particularly difficult to accurately interpret in the presence of low-cardiac output, non-pulsatile flow or collaterals [161]. Patients with Fontan circulation (both HLHS and non-HLHS) have significantly worse post-OHT survival if protein-losing enteropathy is present conferring a 40% early mortality [161,162]. Autopsy studies have shown that Fontan patients have adverse pulmonary vascular remodelling, with the degree of changes correlating with time since Fontan completion [60]. Additional studies have shown that patients long after Fontan surgery have evidence of elevated transpulmonary gradients post heart transplantation [163]. While Fontan patients may not meet diagnostic criteria for PH, the aggressive use of pulmonary vasodilators post-transplant may be considered [164].

RCM is characterized by impaired diastolic function with relatively preserved systolic function. The restriction of filling results in cardiac failure and raised pulmonary vascular resistance. Although some children will remain well for many years, a previous report on children with RCM showed progression to a low cardiac output state with left ventricular failure leading to death or transplantation within 2 years [165]. The rate at which pulmonary vascular disease develops in paediatric RCM is faster than in dilated cardiomyopathy, requiring transplantation at a younger age [166]. Comparing the incidence of preoperative PHT between CHD, DCMP and RCM PH was most severe in the RCM group. Although thirty-day survival was almost equal in all groups (92.0%, 97.1% and 100% for patients with CHD, DCM and RCM, respectively), the incidence of graft-RV-failure was highest for patients with RCM (43.5%; versus CHD, 26.0%; versus DCM, 14.7%). One-year survival estimates for patients with CHD, DCM and RCM were 92.0%, 97.8% and 82.6%, respectively [167].

HLTX in the case of left heart failure is the ultimate procedure in patients with TPG above 15 mmHg and PVR more than 4 Wood Units [51,168], despite aggressive treatment including long term MCS. However, we have to keep in mind that HLTX eliminates the potential opportunity for three other patients on the waitlist to receive an organ (one heart and two lungs) and represents one of the greatest challenges of careful resource utilization. Thus, early consideration for listing to HTX prior to irreversible vascular damage should be the main treatment target.

## 5. Group 3 PH (Table 1)

### Bronchopulmonary Dysplasia (BPD)

Within group 3 PH, developmental lung disorders (Table 1) are the most common causes of paediatric PH, with a prevalence between 10 and 34%, according to registry data and epidemiological studies [1,19,98]. Among these, bronchopulmonary dysplasia (BPD) associated PH comprises an increasingly important entity. BPD is the lung disease of the preterm infant in need of mechanical ventilation and/or oxygen therapy due to respiratory distress syndrome. Today’s (post-surfactant era) “new” BPD occurs in more immature infants born at 24–28 weeks post menstrual age (PMA), weighing ≤ 1000 g. As a result of arrested lung growth ‘‘new’’ BPD is marked by less airway damage but impaired alveolarization, together with disturbed vasculogenesis and vascular simplification and remodelling [169,170]. The decreased vessel density throughout the pulmonary microcapillary network leads to decreased cross sectional area for pulmonary blood flow and increased pulmonary vascular resistance [169,171]. Risk factors for BPD-PH are low birth weight, maternal abnormalities such as oligohydramnion, pre-eclampsia and postnatal injuries, including hyperoxia, hypoxia and infection, all of which lead to endothelial injury, disruption of growth factor expression and the activation of signalling pathways promoting vascular remodelling [169,172,173,174,175]. Genetic susceptibility appears to play a further role in this process. Additional cardiovascular abnormalities frequently seen in preterm infants are persistent patent ductus arteriosus (PDA), aortopulmonary collaterals, interatrial shunts, or pulmonary vein stenoses and predispose infants to increased risk. The prevalence of PH in BPD is estimated between 3% and 44%, depending on PH definition and time of diagnosis with the highest incidence rates in severe BPD [176,177,178]. BPD-PH carries a high mortality with reported death rates between 16% and 40% during the first 2 years of life [169,177,179,180]. Therefore, it seems reasonable to aggressively treat these patients with advanced therapies. Several smaller studies and case reports have investigated the use of Sildenafil and Bosentan in BPD [181,182]. Given the frailty of these patients, PH diagnosis was mostly made by echo and therapy was initiated without RHCs. Though currently there is no approval for targeted therapies in BPD-PH, the EPPVDN recommends the use of sildenafil as first line advanced treatment in patients diagnosed with PH by echo (i.e., TRV > 2.5 m/s, TAPSE < z-score-3, PAAT < 70 ms, RV/LV end-systolic ratio > 1.5, LV end-systolic eccentricity index > 1, S/D ratio > 1.4) after the optimization of conventional treatments, such as ventilation, oxygenation and diuretic therapy. RHC should be considered in the case of worsening and is recommended if patients deteriorate after therapy escalation with ERAs in addition to sildenafil [183].

However, it is important to frequently evaluate associated conditions, such as pulmonary vein stenosis, PDA or ASD and diastolic dysfunction, prior to therapy escalation, as these conditions might make early invasive diagnostic and/or treatment necessary.

Despite the high mortality rate in early life, BPD-PH is likely to resolve over time. A recently published analysis in 28 BPD-PH patients demonstrated that PH mostly resolved at 2.5 years in patients surviving beyond a corrected age of 6 months. On the other hand, there are reports on PA pressure elevation in former BPD patients, diagnosed at school age or in adolescence [184,185,186]. Currently, larger prospective outcome studies to assess the further course of this disease are lacking.

The respiratory aspect of BPD has been linked to COPD in adulthood based on several longitudinal studies demonstrating the ongoing impairment of lung function of BPD survivors lasting to adulthood, supporting the concept of BPD as a precursor of COPD in adulthood [187,188].

Currently, BPD-PH does not play a significant role in adult Group III-PH. However, with improved neonatal intensive care medicine, the increased awareness of this disease and advanced treatments will lead to an increasing number of extremely low birth weight infants surviving with BPD-PH. Prospective outcome trials will show whether or not and to which extent this entity is likely to present in adolescence and adulthood.

## 6. Group 4 PH (Table 1)

### Chronic Thromboembolic Pulmonary Hypertension

Chronic thromboembolic pulmonary hypertension (CTEPH) is very rare in childhood with 0–1.4% of reported cases found in registries [1,102]. Because CTEPH is primarily a disease of advanced age and is pathophysiological, and clinical understanding for CTEPH derives mainly from adult studies. Risk factors are splenectomy, ventriculo–atrial (VA)-shunt and chronic inflammatory states, such as osteomyelitis, inflammatory bowel disease, a history of malignancy and thyroid replacement therapy [189,190].

Among plasma prothrombotic factors, elevated levels of factor VIII and antiphospholipid antibodies are the most important abnormalities associated with adult CTEPH [190]. Paediatric data are scarce and mainly restricted to case reports. One of the larger studies, including 17 paediatric patients, identified an underlying thrombophilic disorder in 68% of patients, which is in contrast to adult data in which only 20% to 40% are reported to have a contributing hypercoagulable state [191,192,193]. Lupus anticoagulant, anticardiolipin antibodies and protein C deficiency were the most common predisposing thrombophilic states in this paediatric cohort [191]. Therefore, chronically ill children with a diagnosis of one of the aforementioned conditions should be carefully screened for PH in the setting of new PH symptoms. Treatment strategies mainly follow concepts of adult CTEPH, including surgical endarterectomy, which has also been described in children [194], medical and interventional treatments.

## 7. Group 5 PH (Table 1)

### 7.1. Pulmonary Hypertension in Patients with Scimitar Syndrome

Scimitar Syndrome describes a complex congenital anomaly consisting of abnormal drainage of the right sided pulmonary veins into the inferior cava vein, variable degree of hypoplasia of the lung artery and an abnormal supply of the lung by collaterals deriving from the aorta or its main branches [195,196,197,198]. Dextroposition of the heart is commonly found as a result of right sided lung hypoplasia [197,198]. The term Scimitar Syndrome derives from a scimitar shaped shadow along the right sided border of the heart frequently seen on chest X-ray, which is caused by the anomalous vein.

The pulmonary manifestations of Scimitar Syndrome include lobar abnormalities with pulmonary sequestration, which is the presence of a sequestered lobe being separated from the bronchial tree with non-functioning lung tissue. Bronchi may be absent or hypoplastic [197]. Pulmonary artery segments may show variable degrees of hypoplasia with atresia, leading to the closure of the abnormal scimitar vein or the contralateral veins [199]. Its aetiology is most probably attributed to a developmental failure of the lung in early embryology [200].

Overall, it is a rare condition reported with an estimated incidence of 1–3 per 100,000 live births, although the true incidence may be higher, because many patients remain asymptomatic (adult form) [201,202].

The incidence of mild cardiac defects in Scimitar syndrome is high, though less frequently complex pathologies, such as the tetralogy of Fallot or obstruction of the left sided heart structures, are likely to be associated. Symptoms and clinical course are determined by the presence of these defects and the extent of abnormal venous drainage. Whereas the adult variant is normally mild, the infantile form has severe outcomes due to the associated pathologies and requires early surgical/interventional treatment. This situation is complicated by the presence of pulmonary hypertension, which then represents the main determinant of morbidity and mortality. This accounts for infants without associated cardiovascular abnormalities as well [195,201,203]. Pulmonary hypertension is attributed to the multiple potential features described above, of which the following mechanisms are involved:
Increased pulmonary blood flow due to anomalous pulmonary venous connection;Left-to-right shunt due to the aorto-pulmonary collateral flow and associated cardiac defects (VSD, PDA, ASD);Restriction of the pulmonary vascular bed because of pulmonary hypoplasia with subsequent volume overload of the contralateral lung;Pulmonary venous obstruction.

The presence of PH in Scimitar Syndrome has been reported in between 30% and 80% cases [199,204]. While increased PA pressures due to collateral flow or cardiac defects are treated surgically or by coil embolization, the surgical repair of anomalous venous drainage can result in pulmonary vein obstruction, which remains a therapeutic challenge. According to a recently published registry study reviewing 485 patients with Scimitar Syndrome, pulmonary vein stenosis occurred in 25% of surgically treated patients and was correlated with negative outcome, conferring with findings reported from previous studies [199,204]. Finally, hypoplasia of the lung with a consecutive increase in PVR is a common developmental disorder leading to PH and has the potential of improvement due to lung growth [145]. The role of pharmacotherapy to treat patients with lung hypoplasia remains speculative, though the use of vasodilators in lung hypoplasia in Scimitar Syndrome and other entities has been reported in uncontrolled studies [182,199,205,206,207]. Pulmonary vein stenosis must be excluded before using targeted therapies in Scimitar patients. The multiple facets of PH in Scimitar Syndrome highlight the fact that PH in this setting cannot easily be classified. The paediatric taskforce of the sixth WSPH faced this difficulty by including this entity into group 5, which defines PH with unclear or multifactorial mechanisms.

### 7.2. Segmental Pulmonary Hypertension

Segmental PH defines the phenomenon of pulmonary vascular disease presenting in one or more parts of the lung vasculature, whereas other segments remain unaffected. In this disorder, different lung segments may not be equally perfused and receive variable blood flow at different pressures and from different sources.

This feature is a common finding in complex CHD and paediatric cardiologists have been familiar with this entity for a long time as the existence of segmental PH in complex CHD has relevant implications on management strategies. However, the term segmental PH was coined at the fiveth WSPH in 2013, where it was categorized into group V (PH due to multifactorial or unclear mechanisms), though it has more similar features to group I PAH (PAH associated with CHD) and a subcategory of group 4 (PH with congenital pulmonary artery stenosis).

The most common lesion associated with segmental PH is Pulmonary atresia (PA) with VSD and MAPCAS (major aorto-pulmonary collateral arteries) or the extreme type of tetralogy of Fallot. Herein we will focus on this entity, which serves as an example for the complex mechanisms contributing to segmental PH and its clinical implications.

PA with VSD and MAPCAs is a complex lesion with a large morphological variability regarding the sources of pulmonary blood flow. The native pulmonary arteries may be absent, and if present they can be either normal in size or hypoplastic and may show a confluence or not. In patients with hypoplastic lung arteries or the absence of native lung vessels, pulmonary blood supply is maintained by MAPCAS frequently deriving from the descending aorta, which can be variable in size, origin, number and arborisation and morphologic patterns. Operability and postoperative surgical outcomes are determined by RV pressure, which indirectly reflects the integrity of the pulmonary vascular bed [208,209,210]. Whereas primary surgical repair is possible in those with well-developed lung arteries, many patients with diminutive pulmonary arteries and unfavourable pulmonary blood supply need a multistage approach, by which surgical or interventional procedures (i.e., RV-PA conduit, unifocalization, systemic to pulmonary shunt, stenting of the duct) promote pulmonary blood flow and the growth of pulmonary vessels [211].

Especially in patients undergoing the multistage approach there may be several reasons for abnormal pulmonary vascular bed with elevated RV pressures. First, there are technical reasons like PA stenosis after modified BT or central shunt implantation or conduit stenosis. Furthermore, MAPCAS tend to develop variable degrees of stenoses, usually at branching points and junctions of the collaterals with the native pulmonary arteries. These occlusions of vessel lumen are caused by myointimal hyperplasia and usually develop whenever these collaterals are exposed to systemic flow and pressure for longer periods—i.e., in the setting of systemic arterial shunts or when unifocalization is performed late [210,212,213]. On the other hand, unobstructed MAPCAs may transmit systemic pressures to the native pulmonary vessels, leading to shear stress induced pulmonary vascular disease.

Finally, hypoplasia of the native pulmonary arteries remains a limiting factor. Surgical or interventional concepts aim to promote vessel growth by adequate flow, but even if the flow induced growth of the proximal pulmonary arteries can be achieved, the development of the distal pulmonary vessels remains unclear [210,214,215]. In patients with complex cardiac lesions with decreased pulmonary blood flow, factors such as hypoxemia or inadequate blood flow are described to trigger postnatal maladaptation or the underdevelopment of the distal pulmonary vessels. Finally unifocalized MAPCAs and pulmonary arteries can get stenosed in the course of disease, even after surgical repair [213,216,217]. An example of PA/VSD and MAPCAs is shown in Figure 2.

Other less common lesions potentially associated with segmental PH include Truncus arteriosus communis, characterized by a common arterial trunk, giving rise to both the systemic and pulmonary circulation in variable patterns. The absence of PA stenosis leads to the early development of bilateral PH. The presence of stenosis of one PA or its branches lead to segmental PH in the contralateral lung. In hemitruncus arteriosus, a lesion in which one PA is normally connected, but the other one derives from the aorta, segmental PH can develop in the latter due to pressure and volume overload. Both lesions are normally corrected early in infancy; however, if there is unequal blood distribution due to postoperative PA stenosis or the hypoplasia of branch arteries, segmental PH may occur in the contralateral PA [218].

The presence of segmental PH has relevant implications on the outcome and management of patients—this is particularly true in the cases of PA/VSD/MAPCAs. There have been early attempts to modify pulmonary vascular disease by targeted therapies. By now, there are a few publications reporting on beneficial effects of bosentan and sildenafil in children and adults with segmental PH, mostly in the context of PA/VSD and MAPCAs [111,219,220,221]. Certainly, there is the need for the clarification of these results and, currently, there is no recommendation on the use of these agents in this entity. However, given the broad heterogeneity and relatively small number of patients and the impossibility to calculate PVR as a strong hemodynamic endpoint, randomized and adequately powered trials are unlikely to be realistic. In view of the severity of disease and the correlated fate of patients in the presence of segmental PH, it is definitely justified to use these agents on an individual basis.

Our institutional approach in this population is the use of targeted therapies to modify vascular remodelling in combination with proactive catheter interventions addressing anatomical or surgically distinct obstructions with the main target to keep RV pressure as low as possible.

### 7.3. PH in Children with Sickle Cell Disease

Sickle cell disease (SCD) is the most important hemoglobinopathy and affects 20–25 million individuals worldwide—it is estimated that approximately 300,000 new-borns are affected each year, 75% of them in Africa and a high number in India [222]. Pulmonary hypertension is a relevant chronic complication of SCD and has been recognized as a risk factor for mortality in adults [223,224,225,226]. The prevalence of PH based on RHC is between 6 and 10.5% [224,227,228]. The use of tricuspid regurgitation velocity (TRV) ≥ 2.5 m/s as a non-invasive surrogate marker for elevated PA pressure reveals a 30% prevalence of PH in adults, with 8–10% of them having more severe PH suggested by a TRV > 3 m/s [174,178]. A similar prevalence of PH is reported in children with an average of 30% (8–66%) with TRV > 2.5 m/s and 8% (4–14%) with a TRV > 3 m/s [229,230,231]. In adults, TRV > 2.5 correlates with increased 3 years mortality and a probability of 50% of death in TRV > 3 m/s [223,227,231], but elevated TRV does not lead to a higher mortality in children [231,232,233]. Increased TVR has been linked to a reduced exercise capacity in children [234].

PH in the context of SCD may present either as precapillary PH, as postcapillary PH, as combined pre- and postcapillary, or as CTEPH [8,223,227]. Underlying mechanisms are determined by the nature of hematological disorders and include hemolysis and its consequences, chronic anemia, leading to high cardiac output, and/or a hypercoagulable state. Intravascular hemolysis leads to a release of free haemoglobin and arginase I, which both cause the depletion of NO and its precursor arginin, resulting in decreased NO signalling and impaired endothelial function. Free haemoglobin catalyzes reactive oxygen species, which lead to vasoconstriction and inflammation [235,236,237]. In addition, the activation of platelets and the coagulation cascade caused by hemolysis promotes a pro-thrombotic state [238,239]. Interestingly patients with SCD are prone to develop in situ thrombosis at the level of the small pulmonary vessels from recurrent vaso-occlusive crisis [240]. Chronic hypoxia and the upregulation of hypoxic responses [223,241,242] (placental growth factor, HIF) might play an additional role in the process, leading to vascular remodelling, as seen in precapillary PH. The pathophysiological mechanisms for CTEPH are attributed to the hypercoagulable state and the associated risks for thrombotic complications, including pulmonary embolism. Besides, auto-splenectomy, as seen in many SCD patients, is known to be a risk factor for CTEPH. Differential diagnosis of CTEPH requires angiography and/or a V/Q scan. Scintigraphic evidence, suggestive of CTEPH, is observed in approximately 12% of SCD patients with pulmonary hypertension [240].

Postcapillary PH may result from the high cardiac output state, which leads to the enlargement of left sided heart chambers and diastolic ventricular dysfunction with an increase in LVEDP or PAWP. Iron overload, resulting in iron deposition into the myocardium, may contribute to the restrictive physiology reported in children and adults [243]. Given the multiple variants of PH in SCD and its different implications on management, echocardiographic assessment remains a useful screening tool for PH but is not satisfactory for accurate diagnosis and classification. For example, postcapillary PH, which accounts for about 50% of PH cases, cannot be diagnosed by echo, as it requires the invasive measurement of wedge pressure. Secondly, TRV can be overestimated due to the high cardiac output state secondary to anaemia. The limited sensitivity of the echo-based estimation of PH is illustrated in several studies, which demonstrated that the positive predictive value for PH ranges between 25% and 39% in patients with TRV 2.5–3 m/s and improves to 66–77% when a threshold of TRV > 3 m/s is used [225,244]. The American Thoracic Society, endorsed by the Pulmonary Hypertension Association and the American College of Chest Physicians therefore recommends RHC in adults and children with TRV > 2.9 m/s or TRV 2.5–2.9 m/s with increased BNP levels or reduced exercise capacity—otherwise echo screening every 1–3 years. For children, there exists no recommendation for or against screening due to a lack of evidence [245].

Treatment modalities with regard to PH in SCD focus on the reduction in hemolysis and associated comorbidities, such as hypoxia, anaemia, vaso-occlusive and acute chest syndrome events. Successful pulmonary endarterectomy has been reported in SCD patients with CTEPH [246,247,248]. The use of targeted therapies has been reported in some smaller studies and case series mainly in adults with promising results [249,250,251,252]. However, a large multicentre trial on Sildenafil in SCD, including adults and children, had to be terminated due to adverse events that were mainly vaso-occlusive pain crises [253]. According to the current guidelines, there is only a weak recommendation for treating selected patients with PAH SCD confirmed by RHC with a marked elevation of PVR and normal wedge pressure. It is recommended to use either PGIs or ERAs. Due to the lack of data in children, therapeutic recommendations are mainly based on expert opinion. According to the consensus statement of the EPPVDN, the use of advanced treatments, especially sildenafil, is considered to have potentially harmful effects due to the reported vaso-occlusive crises (class of recommendation III) [17].

As PH in SCD is now categorized into group 5 of PH, the use of targeted therapies is not approved and should be restricted to PH expert centres, ideally within clinical trials.

## 8. Summary

There has been great progress in the understanding of pathophysiological mechanisms and management in paediatric PH over the past decades. Many aspects are derived from research achievements originating from adult PH, which has provided substantial knowledge and evidence, especially in management modalities. However, paediatric PH has many elements that completely differ from adults. For example, growing from neonatal age to adolescence implies a number of physiological hemodynamic changes and pathophysiological conditions, during which paediatric PH can occur. This issue has been increasingly addressed by the paediatric PVRI taskforce, which has issued specific recommendations within the current guidelines with respect to diagnosis, classification and management in children. Some paediatric aspects as discussed in this review are difficult and not yet resolved and cannot simply be extrapolated from adult data. The present overview provides insights to the current knowledge in order to improve the understanding and management of some of the difficult and rare entities in paediatric PH.

## Figures and Tables

**Figure 1 medicina-56-00420-f001:**
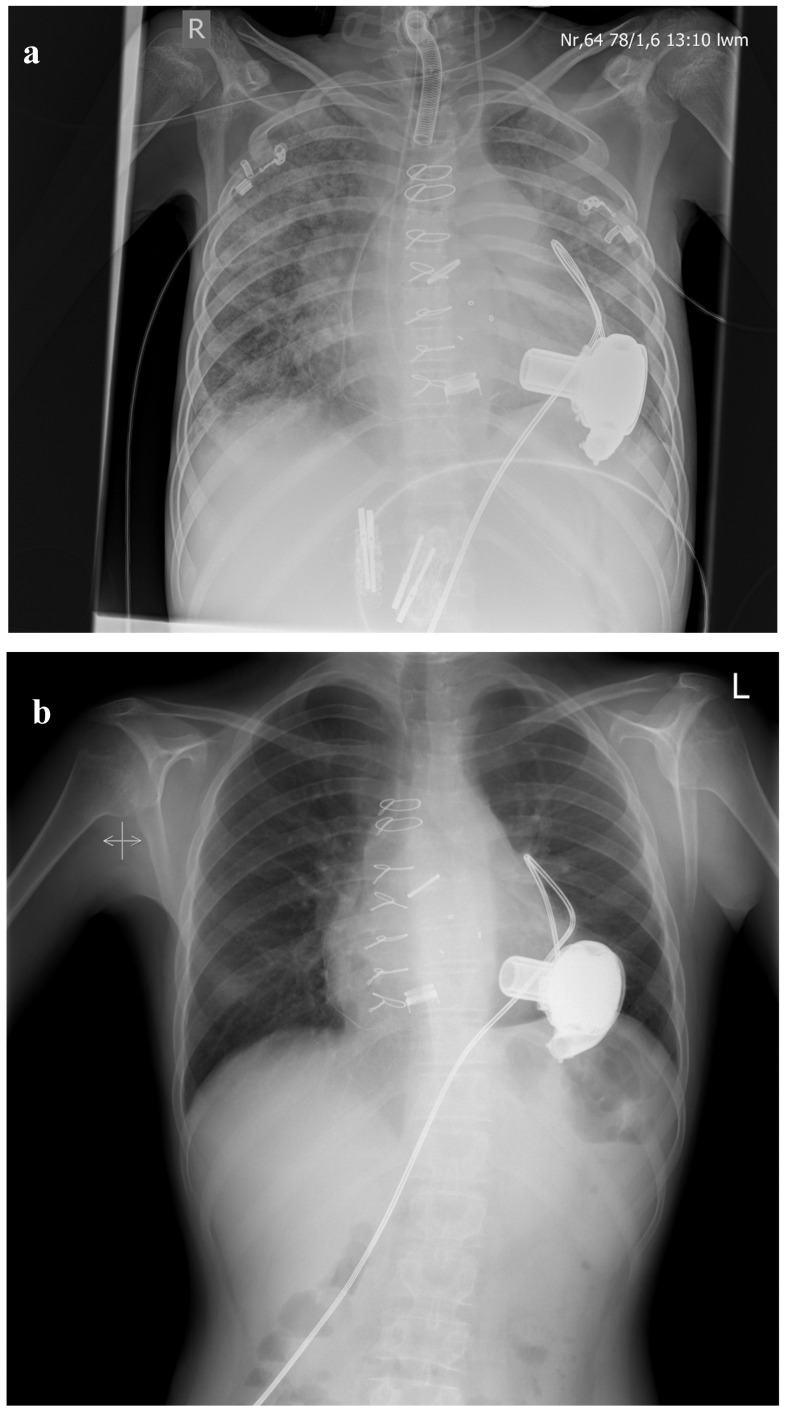
(**a**) Chest X-ray immediately after LVAD-implantation. Fifteen-year-old male with congenital critical aortic valve stenosis after several surgical and interventional procedures (balloon valvuloplasty, 2 × aortic homograft, mechanical aortic valve (St. Jude 17 mm), mitral valve replacement (Mosaic 25 mm)) presenting with severe pulmonary hypertension (RAP mean 20 mmHG, PAP mean 58 mmHG, PAWP 23 mmHG, PVRI 20.2 WU × m^2^). (**b**) Chest X-ray 24 months after LVAD-implantation Pulmonary hypertension resolved (RAP mean 8 mmHG, PAP mean 18 mmHG, PAWP 10 mmHG, PVRI 1.3 WU × m^2^), and successful HTX was performed after 36 months on LVAD.

**Figure 2 medicina-56-00420-f002:**
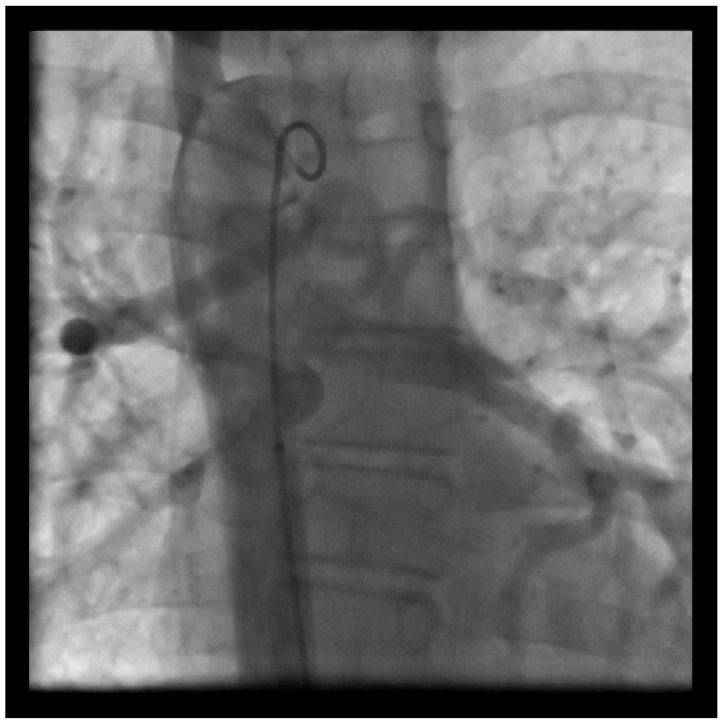
Fifteen-year-old male with Pulmonary Atresia with VSD and MAPCAs (major aortopulmonary collateral arteries) and absence of native pulmonary arteries. Angiogram of descending aorta showing several MAPCAs deriving from the descending aorta, segmental pulmonary perfusion, stenosis to the right lower lobe, pulmonary hypertensive vessels to the left lower lobe.

**Table 1 medicina-56-00420-t001:** Proposal of harmonized hemodynamic and clinical definitions of paediatric pulmonary hypertension.

2019 Hemodynamic Definition of Paediatric Pulmonary Hypertension, Proposed by the European Paediatric Pulmonary Vascular Disease Network [17]		ERS/ESC Updated Clinical Classification of Pulmonary Hypertension (PH) [8]		
Pulmonary Hypertension (PH) mPAP > 20 mmHg—beyond the age of 3 months at sea level				
Precapillary PH (PAH), or pulmonary hypertensive vascular disease (PHVD)	mPAP > 20 mmHg PAWP or LVEDP ≤ 15 mmHg PVR index ≥ 3 WU × m^2^ (DPG ≥ 7 mmHg)	1. Pulmonary Arterial Hypertension 1.1. Idiopathic PAH 1.2. Heritable PAH 1.3. Drug- and toxin-induced PAH 1.4. PAH associated with: 1.4.1. Connective tissue disease 1.4.2. HIV infection 1.4.3. Portal hypertension 1.4.4. Congenital heart disease, including ES - CHD with biventricular circulation mPAP > 20 mmHg and PVR Index ≥ 3 WU × m^2^ - CHD with univentricular circulation (i.e., Fontan) mean TPG > 6 mmHg or PVR index > 3 WU × m^2^ 1.4.5. Schistosomiasis 1.5. PAH long-term responders to calcium channel blockers 1.6. PAH with overt features of venous/capillaries (PVOD/PCH) involvement 1.7. Persistent PH of the new-born syndrome 5. PH with unclear and/or multifactorial mechanisms 5.1. Hematological disorders 5.2. Systemic and metabolic disorders 5.3. Others 5.4. Complex congenital heart disease	3. PH due to lung diseases and/or hypoxia 3.1. Obstructive lung disease 3.2. Restrictive lung disease 3.3. Other lung disease with mixed restrictive/obstructive pattern 3.4. Hypoxia without lung disease 3.5. Developmental lung disorders	4. PH due to pulmonary artery obstructions 4.1. Chronic thromboembolic PH 4.2. Other pulmonary artery obstructions
Isolated postcapillary PH	mPAP > 20 mmHg PAWP or LVEDP > 15 mmHg PVR Index < 3 WU × m^2^	2. PH due to left heart disease 2.1. PH due to heart failure with preserved left ventricular ejection fraction 2.2. PH due to heart failure with reduced left ventricular ejection fraction 2.3. Valvular heart disease 2.4. Congenital/acquired cardiovascular conditions leading to post-capillary PH		
Combined pre- and postcapillary PH	mPAP > 20 mmHg PAWP oder LVEDP > 15 mmHg PVR Index ≥ 3 WU × m^2^ (DPG ≥ 7 mmHg)			

Topics in focus of the article are highlighted blue. Abbreviations: DPG = diastolic pulmonary pressure gradient, LVEDP = left ventricular end-diastolic pressure; PVOD = pulmonary veno-occlusive disease, PCH = pulmonary capillary hemangiomatosis.

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
