# Peer review of "Focused Update on Pulmonary Hypertension in Children—Selected Topics of Interest for the Adult Cardiologist"

_medicina, 2020, doi:10.3390/medicina56090420_

Round 1

Reviewer 1 Report

I have read this review with great interest. It elegantly presents so heterogeneous subject as PAH in children. It has a very high educational value and therefore is worth considering for publication after minor revision. Especially supplementing with current paediatric PAH registers is necessary.

In "Introduction" Line 5: please provide the references for other Paediatric PAH Registries from other countries (such as: France, Poland etc.).

Some editorial and English language corrections are needed before publication. 

Reviewer 2 Report

The authors performed this review to present novel trends and challenges in management of children with pulmonary hypertension (PH), selecting topics seemed to be interesting for adult cardiologists. They discuss 3 groups of PH (according to clinical classification):  group 1 - pulmonary arterial hypertension (PAH), group 2 - PH due to left-sided heart disease and group 5- PH with unclear or multifactorial mechanisms. Revised  definition (introduced by 6th WSPH PH) was clearly discussed. Regarding PAH the authors presented selected issues like an acute vasoreactivity test, pulmonary hemodynamics in Fontan circulation, Eisenmenger syndrome and PH in patients in Down syndrome. In group 2 PH they focused on advanced treatment as mechanical support, and heart or heart/lung transplantation. Group 5 includes miscellaneous condition and authors described a few of them – sickle cell disease, segmental PH, scimitar syndrome.

An advantage of the publication is the part about PAH. Important issues in pediatric cardiology were selected and discussed. A new definition of PH was described, pro and contra regarding it application for children population were clearly presented. Particularly noteworthy is the extensive presentation of pulmonary hemodynamics in Fontan circulation and discussion of use of targeted therapies approved for PAH treatment in Fontan patients. In a short paragraph regarding Down syndrome contribution of various causes (not only CHD) in PH development in these children was highlighted. 6th WSPH also updated pediatric treatment algorithm. Atrial septostomy and reversed Potts shunt was proposed as a palliative procedures in patients refractory to medical treatment to improve symptoms and quality of life as a bridge to transplant. In my opinion it should be mentioned in this part of article because these procedures are rare or even no performed in adults.

In 3rd part regarding PH with unclear or multifactorial mechanisms, the authors presented 3 rare but challenging reasons for pulmonary hypertension. These conditions are not covered in recommendations and require a patient-based approach. The authors clearly described factors contributing to PH development and treatment strategies base on recent publications.

PH due to left-sided heart disease is the most common form of PH in adults, but in children is rare and poorly described in the literature. In adults, postcapillary PH develops due to heart failure with preserved EF. There are limited data on the prevalence and implications of pulmonary hypertension in children and adolescents with advanced left heart failure. The authors briefly described treatment of heart failure. However, this is not a PH treatment, it rather alleviate the symptoms. Described in detail mechanical support and heart transplantation can prevent PH development or even lead to its regression, but they are not widely available in children. On the other hand, authors didn’t discuss the trials of treatment with PH targeted medications in selected cases (with precapillary component?) described in the literature, or palliative atrioseptostomy as a bridge to transplant. The topic of the following part suggests that it is dedicated to the transplantations in PH due to left heart disease. In fact, authors discussed various indications to the heart or heart and lung transplantation (included for instance Eisenmenger syndrome or failing Fontan) not focusing on postcapillary PH. In this part, many interesting publications were cited but I cannot find the keynote ordering the presented information.

The choice of topics is subjective, but in my opinion the group of growing importance in pediatric PH - PH due to lung diseases and/or hypoxia should be mentioned in such review. According to recent guidelines PH in broncho-pulmonary dysplasia can be diagnosed without RHC and treated with targeted PH medicines and treatment discontinuation is possible (see references 154). This strategy is different than management in adults so it seems to be interesting for adult cardiologist.

In summary, the article is very interesting and introduced many new information regarding PAH and multifactorial PH wide documented in literature. While the part about PH due to left-sided heart disease should be completely reworded and requires completion. This is a very difficult subject even for dedicated specialists. In my opinion, it would be better to replace that part with part concerning group 3 which becomes a growing problem for PH specialists. Instead of part regarding transplantation in PH group 2 I suggest to add separated chapter concerning heart or lung or both organ transplantation in various form of PH. In this part also palliative procedures proposed as a bridge to transplant could be discuss.

PH due to left-sided heart disease becomes a more and more important topic in pediatric cardiology, but till now it is not well described. Our knowledge bases mostly on trials in adults or case reports and expert’s experience. It would be valuable for a review article on this topic to be prepared in separate publication.    

Specific comments (line numbers are not available in my copy)

Page 5: … Similar to adults a higher risk in children includes hemodynamic measures such as RAP > 10 mmHg, PVRI >20WU x m2 and mPAP/sPAP ratio > 0.75. Clinical and echocardiographic evidence of right heart failure, progressions of symptoms, WHO functional class III and IV, failure to thrive and elevated or increasing BNP / NT – pro BNP levels are further determinants of high risk in children.

I think all the criteria recommended by the 6th WSPH should be listed, maybe in the table. It is not clear to me why the above were chosen.

In 2019 new criteria were released (see references 23), maybe it will be better to present newest ones.

Page 7 PVR may be difficult to assess because of collateral flow

Such diagnostics can be performed using MRI and RHC together.

Page 10: Currently, there are limited data on the prevalence and implications of pulmonary hypertension in pediatric patients with advanced left heart failure. According to the PH registry in the Netherlands PH due to left heart disease accounts for 5 % of all pediatric PH

Page 10: Figure I illustrates morphological changes on chest x ray before, during and 24 months after treatment with LVAD.

There is no added chest x-ray before LVAD implantation. It could be very interesting.

Page 14 Especially patients undergoing the multistage approach can have several reasons for abnormal pulmonary vascular bed with segmental PH and therefore elevated RV pressures. First, there are technical reasons like PA stenosis after modified BT or central shunt implantation or conduit stenosis. Furthermore, MAPCAS tend to develop variable degree of stenoses, usually at branching points and junctions of the collaterals with the native pulmonary arteries. These occlusions of vessel lumen are caused by myointimal hyperplasia and usually develop whenever these collaterals are exposed to systemic flow and pressure for longer periods, i.e. in the setting of systemic arterial shunts or when unifocalization is performed late (158)(160)(161).

PA stenosis is not a reason of PH (segmental or not), it can cause elevated RV pressure, but not PH. The same MAPCAS stenoses.

Round 2

Reviewer 2 Report

I have no comments. Congratulation, great job.

Author Response

We thank the reviewer for the suggestions and comments. The constructive inputs have contributed to the quality of our work.